# Rare Variants in Primary Immunodeficiency Genes and Their Functional Partners in Severe COVID-19

**DOI:** 10.3390/biom13091380

**Published:** 2023-09-12

**Authors:** Maryam B. Khadzhieva, Dmitry S. Kolobkov, Darya A. Kashatnikova, Alesya S. Gracheva, Ivan V. Redkin, Artem N. Kuzovlev, Lyubov E. Salnikova

**Affiliations:** 1The Laboratory of Clinical Pathophysiology of Critical Conditions, Federal Research and Clinical Center of Intensive Care Medicine and Rehabilitology, 107031 Moscow, Russia; m.had@mail.ru (M.B.K.); palesa@yandex.ru (A.S.G.); artem_kuzovlev@mail.ru (A.N.K.); 2The Laboratory of Ecological Genetics, Vavilov Institute of General Genetics, Russian Academy of Sciences, 119991 Moscow, Russia; dmitry.s.kolobkov@gmail.com (D.S.K.); daria_sv11@mail.ru (D.A.K.); 3The Laboratory of Molecular Immunology, National Research Center of Pediatric Hematology, Oncology and Immunology, 117997 Moscow, Russia; 4The Department of Population Genetics, Vavilov Institute of General Genetics, Russian Academy of Sciences, 119991 Moscow, Russia; 5Competence Center for the Development of AI Technology, Federal Research and Clinical Center of Intensive Care Medicine and Rehabilitology, 107031 Moscow, Russia; iredkin@fnkcrr.ru

**Keywords:** severe COVID-19, primary immunodeficiency (PID) genes, protein-protein interaction (PPI), functional partners (FPs) of PID genes, omnigenic model, core and peripheral genes, whole-exome sequencing, rare high-impact (HI) variants

## Abstract

The development of severe COVID-19, which is a complex multisystem disease, is thought to be associated with many genes whose action is modulated by numerous environmental and genetic factors. In this study, we focused on the ideas of the omnigenic model of heritability of complex traits, which assumes that a small number of core genes and a large pool of peripheral genes expressed in disease-relevant tissues contribute to the genetics of complex traits through interconnected networks. We hypothesized that primary immunodeficiency disease (PID) genes may be considered as core genes in severe COVID-19, and their functional partners (FPs) from protein–protein interaction networks may be considered as peripheral near-core genes. We used whole-exome sequencing data from patients aged ≤ 45 years with severe (*n* = 9) and non-severe COVID-19 (*n* = 11), and assessed the cumulative contribution of rare high-impact variants to disease severity. In patients with severe COVID-19, an excess of rare high-impact variants was observed at the whole-exome level, but maximal association signals were detected for PID + FP gene subsets among the genes intolerant to LoF variants, haploinsufficient and essential. Our exploratory study may serve as a model for new directions in the research of host genetics in severe COVID-19.

## 1. Introduction

COVID-19, caused by the SARS-CoV-2 virus, remains a global health problem. COVID-19 has a variety of clinical manifestations, ranging from asymptomatic infection to fatal respiratory or multiorgan failure. The main risk factors for the severe course of COVID-19 are older age [1], male gender [2], the presence of specific comorbidities and overall multimorbidity [3,4], race/ethnicity and social position [5]. An increased risk of developing severe COVID-19 outcomes is also associated with unhealthy lifestyle factors and an unfavorable socioeconomic status. Low socioeconomic status is often the cause of a lack of timely, quality medical care and unhealthy behaviors. The latter include an unbalanced high-calorie diet, reduced physical activity, increased alcohol and tobacco use, and sleep disturbances. All of these behaviors can weaken a person’s immune status [6,7]. The severity of the course of COVID-19 is also influenced by genetically mediated immune system functionality. Genetic and non-genetic factors can interact to orchestrate a complex cascade of immune signals that lead to the recovery or failure of critical organ systems and death [8,9,10,11,12].

Rare disease findings can help identify causative genes and mechanisms that explain the predisposition and course of some common diseases, and in the case of COVID-19, such rare genetic diseases may be primary immunodeficiencies (PIDs). A recent review including data on 459 PID patients with COVID-19 showed higher mortality, hospitalization rates, and a higher frequency of oxygen supplementation use in PID patients than in the general population [13], but not all PID patients have a severe disease course [14,15]. This is linked to other risk factors, the same as in the general population [15], as well as the severity of the causative variant [16] and the overall genetic background. Based on ideas from the omnigenic model of the heritability of complex traits, we recently showed that the cumulative effect of rare high-impact (HI) genetic variations across the exome is associated with the severity of COVID-19 [17]. The omnigenic hypothesis suggests that genes with regulatory variants in at least one disease-associated tissue may influence the overall risk of disease development [18]. Due to the interconnectedness of gene regulatory networks, association signals from so-called peripheral genes are transmitted to core genes, which have a direct effect on the phenotype. In our previous study, the total contribution of rare potentially pathogenic variants to the phenotype of severe COVID-19 was greater for PID genes than for other analyzed groups of genes potentially important in the context of severe infection development. We hypothesized that PID genes are enriched for core genes for a phenotype defined as severe COVID-19 [17].

Phenotype determination is one of the “open” problems in the genetics of complex human traits because there is often a lot of noise in the phenotyping process (mostly regarding binary phenotypes) [19]. This problem becomes even more relevant when COVID-19 comparison groups are formed according to disease severity, as disease severity is a continuum [20] and it is not always possible to distinguish between moderate and severe forms of disease. Because of the important role of the patient’s immune responses in the pathophysiology of COVID-19, various immunologic tests have been proposed to identify patients at high risk of progression or death caused by COVID-19 [21,22]. One promising immunological test is to measure the levels of T-cell receptor excision circles (TRECs) and kappa-removal recombination excision circles (KRECs), which are non-replicating DNA fragments that are formed during the maturation of T- and B-lymphocytes, respectively, and are retained in cells. Therefore, TRECs and KRECs are considered proxy markers of the emergence of new T- and B-lymphocytes, and their age-adjusted levels can be used to assess the efficacy or dysfunction of the respective branches of cellular immunity. The interest in evaluating TREC and KREC levels in severe COVID-19 is bidirectional. On the one hand, given the demonstrated association between TREC and, to a lesser extent, KREC levels and the development of critical conditions and adverse outcomes in patients with COVID-19 [23,24,25,26] and non-COVID-19 pneumonia [27], TREC/KREC measurements may be useful in order to confirm the classification of COVID-19 severity. On the other hand, given the importance of selecting immunologic tests with high prognostic value, the additional evaluation of TREC/KREC levels in severe COVID-19 may contribute to the selection and subsequent clinical implementation of prognostic tools for patients with COVID-19. 

In the present study, we aimed to further address the hypothesis of a possible role of PID genes as potential core genes for severe COVID-19. According to the omnigenic model, the influence of a peripheral gene on the phenotype is realized through a peripheral network [28]. It can be assumed that the closer a peripheral gene is to the core gene in the tissue regulatory network, the more influence on the phenotype it can exert through the core gene. Under this assumption, genes whose products participate in networks of protein–protein interaction (PPI) with PID gene products may contribute to PID gene association signals, with the magnitude of the effect depending on the confidence of the interaction. Thus, our first objective was to create and analyze sets of genes encoding products involved in interactions with PID gene products with varying degrees of confidence to then assess the cumulative effect of rare HI variants in these genes in severe COVID-19. Since severe COVID-19 is associated with older age and age-related diseases in the general population, for this study, to reduce the possible influence of confounders, young adults (20 individuals) were selected from our previous sample [17] for whom TREC/KREC analysis was performed. Therefore, our second objective was to investigate the levels and prognostic efficacy of TREC/KREC counts in patients with different severities of COVID-19, and to compare TREC and KREC levels with the number of rare potentially pathogenic variants in genes controlling PPI with PID genes.

## 2. Materials and Methods

### 2.1. Study Design

Our study comprised two phases: theoretical and experimental. In the theoretical phase, we compiled and analyzed lists of genes associated with primary immunodeficiencies (PIDs) and their functional partners from PPI networks (hereinafter referred to as FPs). In the experimental phase, we used the results of the whole-exome sequencing analysis in COVID-19 patients with different disease severities to analyze the distribution of rare HI and potentially pathogenic missense variants in the gene sets generated in the previous theoretical phase of the study and based on the findings of this previous phase.

### 2.2. Theoretical Phase: Gene List Construction

The list of primary immunodeficiency genes (PIDs) was taken from the International Union of Immunological Societies (IUIS) Committee of Experts 2022 Updated Classification [29]. The phenotypes and associated genes are presented in this paper in 10 tables, of which Tables I through IX include the phenotype categories associated with germline variants, and Table X includes “Phenocopies.” This latter category was excluded from our analysis. The database search tool (STRING: Search Tool for the Retrieval of Interacting Genes) version 11.5 [30] was used to generate lists of FPs of PID genes. We generated three gene sets: (I) consisting of PID genes, and (II and III) consisting of FPs of PID genes. Sets II and III were generated based on STRING combined scores, which provide an assessment of STRING’s confidence (measured from zero to one) that the putative association between proteins is biologically significant, given all the contributing evidence from different sources. We hypothesized that the magnitude of the effect of FPs would depend on the confidence of their interaction with PID genes, with the greater the confidence, the greater the effect. In set II, FPs had a combined interaction score with PID genes ≥ 0.9, and in set III, the combined interaction score with PID genes ranged from 0.4 to 0.89. STRING was also used for subsequent enrichment analysis. We next compared these three sets of genes for the number of genes that may be biologically important in relation to the development and course of acute infection. As biologically important, we considered the following gene groups: haploinsufficient [31], essential for life [32], intolerant to loss-of-function variants and intolerant to missense variants (https://storage.googleapis.com/gnomadpublic/release/2.1.1/constraint/gnomad.v2.1.1.lof_metrics.by_gene.txt.bgz (accessed on 21 June 2022)), linked to SARS-CoV-2 infection and/or COVID-19 disease from the GENCODE project (https://www.gencodegenes.org/human/covid19_genes.html# (accessed on 12 May 2023)), and immune tissue-specific [33]. Other tissue-specific or non-specific genes were included in the analysis for the comparison with immune tissue-specific genes.

Haploinsufficiency is a prediction of sensitivity to a reduced dose of a gene. DECIPHER presents haploinsufficiency scores that are based on the predicted probability of haploinsufficiency. Scores in the 0–10% range indicate a higher probability that the gene is haploinsufficient (https://www.deciphergenomics.org/about/downloads/data (accessed on 15 September 2022)). Genes intolerant to a loss of function, i.e., HI variants (pLI > 0.9) and missense variants (missense Z score > 3.09), were established based on constraint metrics from gnomAD v2.1 (https://storage.googleapis.com/gnomadpublic/release/2.1.1/constraint/gnomad.v2.1.1.lof_metrics.by_gene.txt.bgz (accessed on 21 June 2022)). The GENCODE project is currently re-annotating genes encoding human proteins associated with SARS-CoV-2 infection and/or COVID-19 disease. From the list of genes under consideration, we have selected those marked as updated. Compiling a list of immune-tissue-specific genes has been described elsewhere [33]. Briefly, we used the TissueEnrich R package v. 1.10.1 [34] to select tissue-enriched, group-enriched and tissue-enhanced genes from the set of RNA HPA tissue gene data (https://www.proteinatlas.org/about/download (accessed on 19 September 2022)). Immune-system-related tissues included appendix, B-cells, bone marrow, dendritic cells, granulocytes, lymph node, monocytes, NK-cells, spleen, T-cells, and tonsil tissues [35].

### 2.3. Experimental Phase

#### 2.3.1. Patients and Clinical Data

Twenty patients with severe (*n* = 9) or mild/moderate COVID-19 (*n* = 11) under 45 years of age from our cohort of 86 patients with whole-exome sequencing data were selected for the study [17]. Patients with COVID-19 were recruited from the M.F. Vladimirsky Moscow Regional Scientific Clinical Institute, the Moscow Clinical Center for Infectious Diseases at Voronovsky, and the V.P. Demikhov City Clinical Hospital of the Moscow Health Department in 2020 during the period preceding the start of vaccination. In addition to age (no older than 45 years), a confirmed SARS-CoV-2 infection test and TREC and KREC data [24] were required for inclusion in the study. Since age-specific TREC and KREC levels differ in patients with immunodeficiency, those taking immunomodulators and opioids, and those with changes in their sex hormone levels [36,37,38], the exclusion criteria in our study were selected to exclude the possible influence of comorbidities, medications, and some health-related conditions (e.g., pregnancy) on the course of COVID-19 and TREC and KREC counts. The exclusion criteria were as follows: patients with incurable terminal illness, primary or acquired immunodeficiency, long-term corticosteroid use, pregnancy, alcoholism, drug addiction, and HIV/AIDS. Although extremely unfavorable socioeconomic conditions and lifestyle were not among the exclusion criteria, such patients were not encountered in our study. For information on clinical diagnosis, see [17].

The Ethics Committee of the Federal Research and Clinical Center of Intensive Care Medicine and Rehabilitology approved the study; all included patients or their legal representatives signed an informed consent form.

#### 2.3.2. Sample Analysis

The TREC/KREC levels were assayed in whole blood samples as described previously [24,27]. In brief, DNA was isolated from 200 μL of venous blood via isopropanol precipitation. The RT–qPCR reactions were performed in a final volume of 25 μL of reaction mixture containing 200 ng of DNA, primers and probes, which were designed for the specific amplification of the δREC–ψJα T-cell receptor, kappa-deleting joint and human albumin (reference gene). The PCR conditions were as follows: 7 min at 95 °C, followed by 45 cycles of 30 s at 93 °C and 1 min at 59 °C (CFX96, Bio-Rad, USA; manufactured in Singapore). Standard curves for the accurate quantification of TREC, KREC, and albumin were obtained by constructing a calibration curve from sequentially diluted genetic constructs containing the TREC/KREC junction region and the corresponding albumin gene region. TREC and KREC copies were calculated and expressed as copies per 100,000 nucleated cells using the following formula: [mean TREC(KREC) value/(mean albumin value/2)] × 100,000.

DNA isolation and sequencing is described elsewhere [17]. Briefly, DNA was isolated from blood using the Qiagen DNA blood mini kit DNA. The Swift 2S^®^ Turbo DNA Library Kit was used for fragmentation and barcoding. Enrichment was performed with the Twist HumanCoreExome (https://www.twistbioscience.com/products/ngs/fixed-panels/human-core-exome (accessed on 29 January 2021)). Sequencing was executed on an Illumina Hiseq X Ten platform with 150 bp paired-end reads. Reads were aligned to the GRCh38 reference genome using BWA MEM [39]. Duplicate reads were marked and excluded using the MarkDuplicates program. Variant calling was carried out using the HaplotypeCaller program of the GATK package. Variants were required to pass GATK’s standard variant quality score recalibration (VQSR) threshold along with additional filters [17], and to have at least 10× coverage.

#### 2.3.3. Annotation of Variants

Variants were annotated using SnpSift [40], SnpEff [41], and FAVOR [42], as well as the population databases Genome Aggregation Database (GnomAD) [43], 1000G [44], and TopMed [45]. Variants assessed by Ensembl as having serious consequences for protein structure and function (acceptor splice variants, donor splice variants, stop-loss, frameshift, stop-loss, and start-loss) were classified as HI. To classify a missense variant as harmful, we used the rare exome variant ensemble learner (REVEL) tool with a recommended threshold > 0.5 [46]. Our analysis focused on rare variants with alternative allele frequency (AF) < 0.001 or no AF data (missing) in the GnomAD, 1000G, and TopMed population resources (hereinafter rare variants).

### 2.4. Data Analysis

Calculations were performed using the R software (version 3.4.1). In the theoretical phase of this work, we compared the proportions of biologically important genes in PID genes (set I) and their FPs with combined interaction scores with PID gene products ≥ 0.9 (set II) and 0.4 to 0.89 (set III). Because the gene sets overlapped, for statistical analysis, we excluded all genes of set I from set II and all genes of sets I + II from set III. Thus, as designed, we categorized genes according to their proximity to the phenotype in terms of causality. The analyzed sets did not contain common genes, and we used Pearson’s chi-square test with Yates’ correction for continuity. In the experimental phase, the calculation of the area under the receiver operating characteristic (ROC) curve (AUC) was performed using the approach of DeLong et al. [47]. The AUC was considered according to Metz [48]; AUC > 0.9 means that the diagnostic performance of the classifier is excellent. Given the sample size, we used the unadjusted two-sided Cochran–Mantel–Haenszel (CMH) test to analyze the sequencing data. The pooled analysis of rare variants was carried out with the dominant inheritance model. Graphs were plotted using https://www.bioinformatics.com.cn/en (accessed on 23 June 2023), a free online platform for data analysis and visualization.

## 3. Results

### 3.1. Theoretical Phase: The Analysis of PID Genes and Their Functional Partners

The list of PID phenotypes and associated genes is constantly growing; for example, in 2004, the IUIS Committee update included 57 gene–phenotype pairs [49], and by 2022, this number had increased to 450 [29]. The IUIS reports categorize phenotypes into tables, which in turn consist of subtables with overlapping phenotypes. Most of the new genes, as exemplified by those established since 2004, encode proteins that are FPs of previously known proteins, so similar phenotypes in the subtables are associated with genes that form PPI networks (Appendix A). Although some other genes that will be identified as PID genes in the future can be expected to also encode proteins involved in PPI networks with previously known FPs, we are interested in finding differences between PID and FP genes today, when NGS technologies are already used, including for patients with PID [50,51].

The three sets of genes we analyzed in this study included (I) 450 PID genes [29], (II) 4580 FPs of PID genes with a combined interaction score of ≥0.9, and (III) 6445 FPs of PID genes with a combined interaction score of 0.4 to 0.89 (Appendix A). Gene ontology and KEGG pathway enrichment analyses were performed for PID genes and their FPs representing similar phenotypes, i.e., belonging to the same IUIS subtables. Because the gene sets overlapped (shown in the legend panel in Figure 1), we considered sets I, I + II, and I + II + III (Figure 1A,B; Appendix A). Although PID genes accounted for a smaller proportion of the genes in sets II and III, the results of the enrichment analyses in these three sets were consistent with each other. These results reflected the predominance of different types of regulatory interactions (Figure 1C,D).

We also compared the proportion of biologically important genes (see Section 2.2) in the three sets under consideration and found that sets I and II did not differ from each other, but differed from set III in the proportion of genes that are haploinsufficient, intolerant to LoF and missense variants, essential, and annotated by GENCODE-COVID-19 (Figure 1E,F; Appendix A). The distribution of immune tissue-specific genes did not correspond to this pattern (Figure 1E,F; Appendix A). The proportion of immune tissue-specific genes was higher in set I than in sets II and III. Set I was depleted in non-immune tissue-specific genes compared to sets II and III. The highest proportion of tissue non-specific genes was reported in set II compared to set III, while the other results were non-significant. Thus, pronounced differences in tissue specificity were observed for PID genes and their FPs.

As follows from the theoretical phase of this work, among the FP genes there is a large share of genes involved in the same biological processes and metabolic pathways as the PID genes themselves. In addition, among the FPs (mostly in set II), as well as among the PID genes themselves, there is a high proportion of biologically important genes whose variability may influence the resistance of the organism in response to acute infection. This hypothesis was tested in the experimental phase of this work.

### 3.2. Experimental Phase; Whole-Exome Sequencing of Twenty Patients with COVID-19 

#### 3.2.1. Demographic and Clinical Characteristics of Patients

Twenty unrelated patients with COVID-19 aged ≤45 years were included in the present study. Key demographic and clinical data are presented in Appendix A. The severe COVID-19 subgroup included nine patients with a severe or extremely severe course (mean age ± SD, 38.33 ± 5.72), and the non-severe COVID-19 subgroup included eleven patients with mild or moderate COVID-19 (35.18 ± 5.91). The subgroups did not differ in demographic characteristics and previous diseases.

#### 3.2.2. TREC and KREC Levels in Severe and Non-Severe COVID-19

A strong difference was found in the number of TRECs (copies/10^5^ cells) in patients with severe (Median; Q1–Q3: 11.41; 3.74–23.50) versus non-severe (96.63; 71.42–192.41) COVID-19; Mann–Whitney U test *p*-value 0.00063 (Figure 2A, Appendix A). The TREC levels did not differ between men and women for severe COVID-19 and were higher in women than in men for non-severe COVID-19, although the results were not significant after adjustment for multiplicity. The KREC levels did not differ between men and women for either severe or non-severe COVID-19 (Appendix A). Other immunologic markers, such as the leukocyte, lymphocyte, neutrophil, and monocyte counts, as well as the neutrophil-to-lymphocyte ratio (NLR) and lymphocyte-to-monocyte ratio (LMR) at admission and at the last measurement before discharge/death, did not differ significantly between patients with different COVID-19 severities (Appendix A).

In the ROC analysis, the TREC assay demonstrated excellent diagnostic performance for severe COVID-19 (AUC 0.96, 95% CI 0.765 to 1.000, *p* < 0.001, Youden index J 0.818) (Figure 2B). The results for KRECs were non-significant (Figure 2A,B; Appendix A). According to the literature, SARS-CoV-2 infection affects both T and B cell generation, but this effect is less pronounced in the B cell compartment [26]; our sample size was apparently insufficient to detect effects at the level of KRECs. The pronounced effects found for TRECs may be related to the fact that SARS-CoV-2 can directly invade the thymus and alter the gene expression patterns of the thymic epithelium [26].

Given the association demonstrated in the literature between TREC levels and the development of critical conditions and adverse outcomes in patients with COVID-19 and non-COVID-19 pneumonia [23,24,25,26,27], we believe that the results of ROC analysis support the accuracy of phenotype determination in severe and non-severe COVID-19. 

#### 3.2.3. Correlation between TREC and KREC Levels and the Number of Rare High-Impact Variants at the Whole-Exome Level

A significant inverse correlation was observed between the number of TRECs (copies/10^5^ cells) and the number of rare HI variants in the study cohort (Figure 2C). The Spearman rank correlation coefficient *r_s_* was −0.53 (two-tailed *p*-value was 0.017).

#### 3.2.4. Rare Variant Burden in PID Genes and Their Functional Partners in Severe COVID-19

In the entire sample of 20 patients, the total number of rare HI variants in PID genes was small (eight variants in seven genes, Appendix A), so our primary analysis (without dividing into specific gene groups) involved comparing the burden of rare HI variants in all genes and in PID + FP genes (sets I + II and I + II + III) between patients with severe and non-severe COVID-19 (Figure 2D). In our sample, the burden of rare HI variants measured according to OR (95% CI) in severe versus non-severe COVID-19 decreased in the sets I + II > I + II + III > all genes, but the results were highly significant for all sets. We then performed a similar analysis for genes intolerant to LoF variants, haploinsufficient, essential, and immune tissue-specific. For genes intolerant to LoF variants and essential genes, the distribution of association signals was consistent with that obtained earlier for all genes (I + II > I + II + III >all genes); for haploinsufficient genes, the greatest effect was observed in the I + II + III set. The association results for immune-tissue-specific genes results did not remain significant after adjusting for multiple comparisons, so the groups of non-immune-tissue-specific or tissue-non-specific genes were not considered (Figure 2D). GENCODE-COVID-19 genes were not considered because they included only six rare HI variants.

For missense variants, the effect had the same direction of association but was weaker than for HI variants, and in the smaller sets (REVEL > 0.5, Z-score > 3.09, and REVEL > 0.5, Z-score > 3.09 in the PID + FP gene sets) the results were non-significant when corrected for multiple comparisons (Figure 2E).

Summarizing these results, we can conclude that there was a burden of rare potentially pathogenic variants at the whole-exome level in patients with severe compared to non-severe COVID-19. The association signals were stronger when considering certain groups of genes, particularly genes intolerant to LoF variants, haploinsufficient and essential genes. The maximum effect sizes were observed in PID + FP gene subsets selected within these gene groups. Expanding the sample (from I + II to I + II + III) led to a decrease in the variant burden in genes intolerant to LoF and essential genes. Notably, among the PID + FP subsets with the largest effect sizes, PID genes accounted for only 13.6% (3/22), 6.7% (1/15 genes), and 15.6% (5/32 genes) among the genes intolerant to LoF variants, haploinsufficient, and essential, respectively (Figure 2D, Appendix A).

## 4. Discussion

In this study, we turned to the omnigenic hypothesis to delineate the role of PID genes as core genes in severe COVID-19 and the FPs of PID genes at the level of PPI networks as near-core peripheral genes. In the theoretical phase, we compared PID genes (set I) and their FPs with combined interaction scores ≥ 0.9 and 0.4 to 0.89 (sets II and III, respectively) according to several characteristics. We found similarities between PID genes and FP genes with respect to their involvement in biological processes and metabolic pathways, and with respect to FPs from set II in the proportion of biologically important genes in the context of overall organismal resilience and resistance to various stressors. The main differences between PID genes and their FPs were related to tissue specificity. In the experimental phase of this work, we combined the TREC/KREC immunologic analysis in COVID-19 patients with the whole-exome sequencing results. Using a cohort of 20 patients under 45 years of age with different severities of COVID-19, we showed the excellent ability of the TREC count (copies/10^5^ cells) to help diagnose severe COVID-19 (AUC = 0.96), as well as an inverse correlation between the number of rare HI variants and the number of TRECs. In this small cohort, the results of the whole-exome sequencing analysis were consistent with those previously described [17], i.e., patients with severe COVID-19 had an excess of rare potentially pathogenic variants at the whole-exome level. The association effects were stronger when specific groups of genes were considered, particularly genes intolerant to LoF variants, haploinsufficient and essential genes, but not immune tissue-specific genes. In most cases, the effects decreased in the following series: set I + II > set I + II + III > all genes in the group or at the whole-exome level. 

The main question linked to the omnigenic model and discussed in the literature is the identification of core and peripheral genes. While core genes are expected to have largely consistent effects, the effects of peripheral genes are mediated through interactions with environmental and genetic factors and thus have inter- and intrapopulation variability [28]. We extended this hypothesis by considering that the magnitude of the effect of peripheral genes may be influenced by their proximity to core genes in regulatory networks due to fewer interactions during associative signal transduction. The results are consistent with our assumption, as the effects of PID + FP genes, among which FP genes strongly dominated, were higher than the total effects of all genes considered, especially within groups of biologically important genes. The possibility of the gradual delineation of peripheral genes has already been mentioned in the literature [52], and our work illustrates an approach to such delineation and the isolation of a near-core “layer” of peripheral genes.

One of the main differences between core and peripheral genes in our study was the significantly lower number of tissue-specific genes among peripheral genes. It has been shown that disease-causing genes are often tissue-specific and, in a healthy state, are expressed at a higher level in those tissues that are affected in pathology [33,53,54]. In this context, one of the possible explanations for why biologically important peripheral genes from the near-core FPs remain peripheral is their low tissue specificity. FP genes are mainly genes that control various types of interactions; among them there are few genes specific to immune tissue, and they are enriched with non-specific genes that control the processes in various tissues.

We showed the utility of TREC analysis in differentiating patients at increased risk of severe COVID-19 and an inverse correlation between TREC level and the number of rare HI variants. This finding may simply reflect the correlation of disease severity with both variables, the number of TRECs and the number of genetic variants. But it is also known that an excess of rare variants in large specific gene groups is associated with the severity of the course, and sometimes, the age of manifestation of a number of psychiatric [55,56], neurologic [57,58], immune [59], and cardiovascular [60] diseases. Given that the immune system is involved to some degree in most pathologic phenotypes [61,62], their more frequent and severe manifestations may lead to earlier immunosenescence, which is reflected in TREC levels.

## 5. Conclusions

The main limitation of the study is the small sample size, which may be responsible for several types of bias. This applies not only to the genetic part of the study, but also to the interpretation of the results of the TREC/KREC analysis due to their high variability and dependence on many demographic, environmental and patient health factors. Therefore, a larger and more diverse cohort is needed to increase the statistical power of the results and to provide a more complete understanding of the observed patterns and their significance for clinical outcomes. Thus, we consider our study to be a preliminary one, which nevertheless has methodological and biological significance. In a sample of young patients with severe and non-severe COVID-19, we confirmed our previous findings regarding the cumulative effect of rare HI variants at the whole-exome level on individuals’ susceptibility to severe COVID-19 [17]. COVID-19 is a complex multisystem disease associated with multiple genes, most of which are peripheral genes whose action is modulated by environmental, epigenetic and genetic factors [28,63]. Focusing on the ideas of the omnigenic model, which divides genes by their proximity in terms of causality to the phenotype, we used PPI networks to identify the sets of FP genes that are closest to PID genes, which were treated as core genes in this study. We hypothesized that the effects of near-core FP genes might be more pronounced and more stable than those of other genes expressed in disease-relevant tissues. Consistent with this assumption, top association signals for the severe form of COVID-19 were obtained for the subsets of PID + FP genes within the groups of biologically important genes (intolerant to LoF variants, haploinsufficient and essential for life). In most cases, the largest effects of rare variants were obtained for the set of PID genes and their FPs with a combined interaction score ≥ 0.9 (set I + II), providing indirect evidence for the role of proximity to core genes in PPI networks. Our work may serve as a model for systematic studies and new directions in the study of host genetics in severe COVID-19.

## Figures and Tables

**Figure 1 biomolecules-13-01380-f001:**
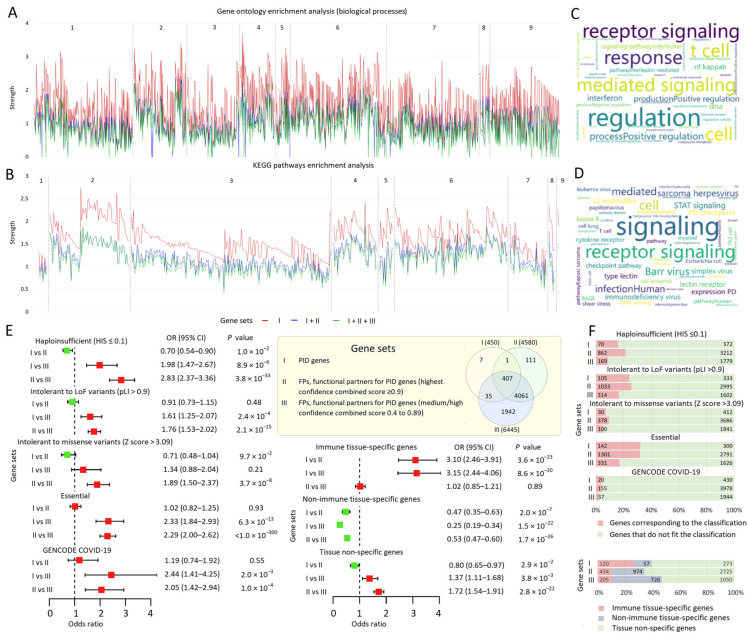
PID genes and their functional partners from the STRING database. (**A**) STRING Gene Ontology and (**B**) KEGG pathway enrichment analysis for sets of genes: (I) PID genes, (II and III) PID genes and their functional partners (FPs) with a combined interaction score ≥ 0.9 and 0.4 to 0.89, respectively. The legend panel for gene sets includes a built-in Venn diagram showing the overlap between these sets and the total number of genes in the sets (indicated in parentheses near each element of the Venn diagram). The analysis was performed for genes (and their partners) representing similar phenotypes, i.e., belonging to the same subtables from the IUIS tables. A list of subtables is provided in Appendix A. For sets II and III, only the terms encountered for the PID genes are indicated. Dotted lines separate enriched terms for genes (and their partners) from each of the IUIS categories, which are signed using Arabic numerals above the line graphs. See Appendix A for more information. (**C**,**D**) Wordclouds displaying sets of terms, related to enriched biological processes (**C**) and metabolic pathways (**D**); the higher the frequency, the larger the word in the wordcloud. (**E**,**F**) A comparison of the proportion of biologically important and tissue-specific genes in gene sets under consideration. Because the gene sets overlapped, we excluded PID genes from set II and PID and set II genes from set 3. (**E**) Odds ratios and horizontal bars denoting 95% confidence intervals are shown. The experiment-wise *p*-value threshold corresponds to 0.0021 to account for multiple testing (0.05/24 comparisons). (**F**) The number of genes from the considered sets is indicated using a 100% stacked bar chart.

**Figure 2 biomolecules-13-01380-f002:**
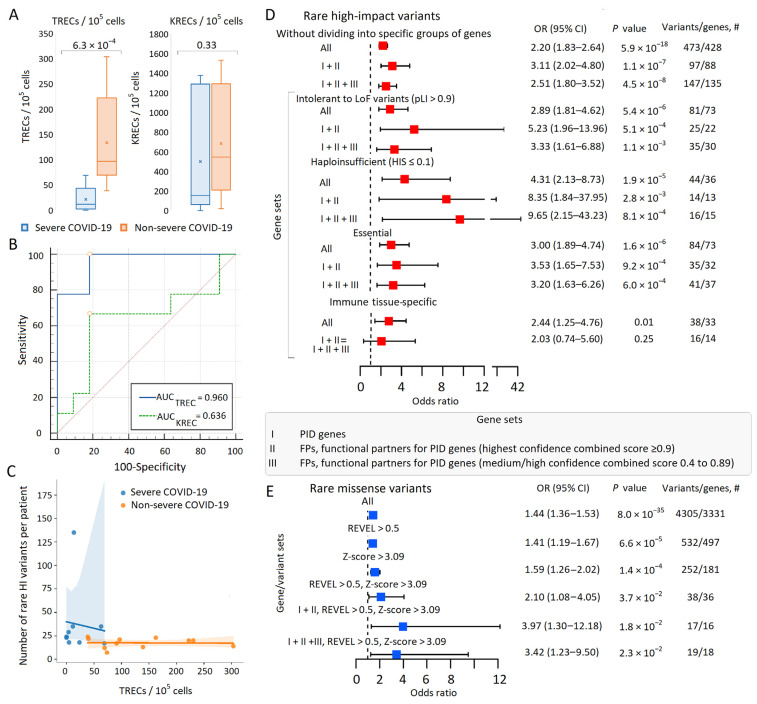
Immunological and genetic variability in patients with severe and non-severe COVID-19. (**A**) Boxplots depicting differences between TREC and KREC levels in patients with severe and non-severe COVID-19. (**B**) Receiver operating characteristic (ROC) curves for TREC and KREC levels in predicting the severity of COVID-19. (**C**) Multi-group scatter plots showing the inverse correlation between TREC levels and the number of rare HI variants in patients with severe and non-severe COVID-19. (**D**,**E**) Burden of rare HI variants (**D**) and missense variants (**E**) in gene sets under study. Odds ratios and horizontal bars indicating 95% confidence intervals are provided. The right column indicates the number of rare variants ((**D**): HI variants; (**E**): missense variants) and the number of genes with these variants in the entire sample of 20 patients. The *p*-value threshold corresponds to 0.0025 to account for multiple testing (0.05/20 comparisons). #: number.

## Data Availability

All raw sequencing data have been submitted to the NCBI BioProject database (https://www.ncbi.nlm.nih.gov/bioproject/ (accessed on 29 June 2023)) under accession number PRJNA947511.

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
