# Peer review of "Rare Variants in Primary Immunodeficiency Genes and Their Functional Partners in Severe COVID-19"

_biomolecules, 2023, doi:10.3390/biom13091380_

Round 1

Reviewer 1 Report

Comment:

The article explores the factors influencing severe COVID-19 outcomes. It focuses on genes associated with immune response and their potential impact on disease severity. The study investigates rare genetic diseases, particularly primary immunodeficiencies (PIDs), and their links to severe COVID-19. Author has also added protein-protein interaction analysis to identify the functional partners (FPs) to deepen the understanding of how PID genes and their FPs collectively influence the immune system's response to infections like severe COVID-19. Additionally, the research examines a promising immunological test involving TREC/KREC levels in severe COVID-19 patients, aiding disease severity classification, prognostic test selection, and insights into immune response and disease progression.
The study investigated the presence of rare potentially pathogenic variants (eight variants in seven genes) in PID genes. Analysis indicated an increased presence of rare potentially pathogenic variants at the whole-exome level among patients with severe patients.

This well-performed study provides a comprehensive exploration into the genetic factors influencing severe COVID-19 outcomes. By applying the omnigenic model and analyzing rare potentially pathogenic variants in PID genes and their functional partners, the researchers uncover novel insights into the interplay of genes and their roles in disease severity. The incorporation of clinical data and the examination of immunological tests further enhance the significance of this research in elucidating the intricate genetic landscape of severe COVID-19. While the study's strengths are evident, I have a few minor comments for further enhancing the clarity and coherence of the manuscript.

Minor Comments:

1.     Could you elaborate the inclusion and exclusion criteria for selection of patient cohort.

2.     Were any gender-specific differences observed in TREC and KREC levels.

3.     While the study presents compelling insights into the association between TREC and KREC levels with COVID-19 severity, it's important to acknowledge the limitation of a relatively small sample size. A larger and more diverse cohort would strengthen the statistical power and robustness of the findings, ensuring a more comprehensive understanding of the observed patterns and their significance in clinical outcomes.

4.     The study's findings regarding TREC levels being significantly elevated in non-severe COVID-19 patients in comparison to those with severe cases suggest an intriguing pattern of immune response variation. However, the lack of significant difference in KREC levels between the two groups raises questions about the role of B-lymphocyte maturation in disease severity. Further exploration into the underlying mechanisms behind these distinctions could shed light on the complex interplay between immune factors and disease progression in COVID-19.

5.     How would the author address potential confounders, such as comorbidities or medications, that might influence TREC and KREC levels in COVID-19 patients and impact the interpretation of the results?  Please add the medication information if possible.

6.     Given the intricate interplay between genetic factors and environmental influences, could the study elaborate on how factors like lifestyle, exposure, or socioeconomic status might contribute to disease severity?

7.     Considering the complexity of immune responses, were other immunological markers or factors, such as cytokine profiles or lymphocyte counts, examined in conjunction with TREC and KREC levels to provide a more comprehensive understanding of the immune dynamics in COVID-19 patients?

8.     Did you look for any Copy number variations (CNVs) or any variants in intron exome boundaries from you Whole exome analysis in the PID or FP genes. As these variants might play an important in any gene function.

9.     On line 412 author mentioned about the string network analysis, where full STRING network was queried for interactors with the maximal available size cutoff of 50 and combined interaction score of ≥0.4.
The incorporation of a combined score includes scores from text mining, chromosome neighborhood, and database annotations that might introduce a potential chance for bias in the selection of functional partners. Could you kindly provide a rationale or justification for using this scoring approach?

Author Response

We are grateful to the reviewer for the constructive generalization of our results, indicating limitations and recommendations. We have done our best to answer all questions. Changes in the text of the manuscript are highlighted in yellow. The line numbers refer to the current (revised) version of the manuscript.

Comment 1. Could you elaborate the inclusion and exclusion criteria for selection of patient cohort?

Response. Following the Reviewer’s comment, we have added this information. Please also refer to our response to comment 6 (lines 169-174).

Lines 167-169. In addition to age (no older than 45 years), a confirmed SARS-CoV-2 infection test and TREC and KREC data [24] were required for inclusion in the study.

Lines 174-178. Exclusion criteria were: patients with incurable terminal illness, primary or acquired immunodeficiency, long-term corticosteroid use, pregnancy, alcoholism, drug addiction, and HIV/AIDS. Although extremely unfavorable socioeconomic conditions and lifestyle were not among the exclusion criteria, such patients were not encountered in our study. For information on clinical diagnosis, see [17].

Comment 2. Were any gender-specific differences observed in TREC and KREC levels?

Response. Thank you for your valuable comment. As per your suggestion, we have considered gender differences. These data have been added to Supplementary Table 5. Because we added some new data (see also our response to comment #7), most of which had non-normal distributions, we changed the presentation of the data from mean ± SD to median (Q1-Q3).

Lines 305-308. TREC levels did not differ between men and women in severe COVID-19 and were higher in women than in men in non-severe COVID-19, although the results were not significant after adjustment for multiplicity. KREC levels did not differ between men and women for either severe or non-severe COVID-19 (Supplementary Table 5).

Comment 3. While the study presents compelling insights into the association between TREC and KREC levels with COVID-19 severity, it's important to acknowledge the limitation of a relatively small sample size. A larger and more diverse cohort would strengthen the statistical power and robustness of the findings, ensuring a more comprehensive understanding of the observed patterns and their significance in clinical outcomes.

Response. Thanks for your comment. We have addressed your concerns.

Lines 429-434. This applies not only to the genetic part of the study, but also to the interpretation of the results of the TREC/KREC analysis due to their high variability and dependence on many demographic, environmental and patient health factors. Therefore, a larger and more diverse cohort is needed to increase the statistical power of the results and to provide a more complete understanding of the observed patterns and their significance for clinical outcomes.

Comment 4. The study's findings regarding TREC levels being significantly elevated in non-severe COVID-19 patients in comparison to those with severe cases suggest an intriguing pattern of immune response variation. However, the lack of significant difference in KREC levels between the two groups raises questions about the role of B-lymphocyte maturation in disease severity. Further exploration into the underlying mechanisms behind these distinctions could shed light on the complex interplay between immune factors and disease progression in COVID-19.

Response. Thank you, we have now added several sentences to address the Reviewer’s comment.

Lines 317-321. According to the literature, SARS-CoV-2 infection affects both T and B cell generation, but this effect is less pronounced in the B cell compartment [26]; our sample size was apparently insufficient to detect effects at the level of KRECs. The pronounced effects found for TRECs may be related to the fact that SARS-CoV-2 can directly invade the thymus and alter the gene expression patterns of the thymic epithelium [26].

Comment 5. How would the author address potential confounders, such as comorbidities or medications, that might influence TREC and KREC levels in COVID-19 patients and impact the interpretation of the results?  Please add the medication information if possible.

Response. Thank you for your comment. We have added some explanations (lines 169-174). We collected information on medications used for the full sample of 86 patients in the previously published study (reference 17). However, this information is missing for the patients from the subsample used in this study, who were younger and healthier than other patients. Given this fact and our rather strict exclusion criteria, we believe that medications could not have significantly influenced our results.

Lines 169-174. Since age-specific TREC and KREC levels differ in patients with immunodeficiency, while taking immunomodulators and opioids, and with changes in sex hormone levels [36,37,38], the exclusion criteria in our study were selected to exclude the possible influence of comorbidities, medications, and some health-related conditions (e.g., pregnancy) on the course of COVID-19 and TREC and KREC counts.

Please also see our response to your comment 1.

Comment 6. Given the intricate interplay between genetic factors and environmental influences, could the study elaborate on how factors like lifestyle, exposure, or socioeconomic status might contribute to disease severity?

Response. This comment has been addressed (lines XX-XX, lines XX-XX, lines XX-XX).

Lines 40-45. An increased risk of developing severe outcomes of COVID-19 is also associated with unhealthy lifestyle factors and unfavorable socioeconomic status. Low socioeconomic status is often a cause of lack of timely, quality medical care and unhealthy behaviors. The latter include an unbalanced, high-calorie diet, reduced physical activity, increased alcohol and tobacco use, and sleep disturbances. All of these behaviors can weaken a person's immune status [6,7].

Lines 47-48. Genetic and non-genetic factors can interact to orchestrate a complex cascade of immune signals that lead to recovery or failure of critical organ systems and death [8,9,10,11,12].

Lines 176-178. Although extremely unfavorable socioeconomic conditions and lifestyle were not among the exclusion criteria, such patients were not encountered in our study.

Comment 7. Considering the complexity of immune responses, were other immunological markers or factors, such as cytokine profiles or lymphocyte counts, examined in conjunction with TREC and KREC levels to provide a more comprehensive understanding of the immune dynamics in COVID-19 patients?

Response. We appreciate your attention to other immunological markers. We have added the available data in Supplementary Table 5. Hematological parameters did not differ between patients with different COVID-19 severity, which is consistent with a higher prognostic value of the TREC assay compared to routine hematological tests.

Lines 308-312. Other immunologic markers, such as leukocyte, lymphocyte, neutrophil, and monocyte counts, as well as neutrophil-to-lymphocyte ratio (NLR) and lymphocyte-to-monocyte ratio (LMR) at admission and at the last measurement before discharge/death, did not differ significantly between patients with different COVID-19 severity (Supplementary Table 5).

Comment 8. Did you look for any Copy number variations (CNVs) or any variants in intron exome boundaries from you Whole exome analysis in the PID or FP genes. As these variants might play an important in any gene function.

Response. Thank you for your interest to our work. We did not screen for CNVs. Our WES data include variants in intron exome boundaries, but in this study, we focused on the cumulative effects of the most promising variants in the context of predicting pathogenicity. Unfortunately, our sample is too small to detect the effects of individual variants. We are currently collecting a sample of critically ill patients and will continue to work on the role of the genetic landscape in the course of acute infections. We hope to collect a larger sample for future studies and hope to be able to look at the effects of individual variants, including those at intronic exome boundaries as you suggested.

Comment 9. On line 412 author mentioned about the string network analysis, where full STRING network was queried for interactors with the maximal available size cutoff of 50 and combined interaction score of ≥0.4.
The incorporation of a combined score includes scores from text mining, chromosome neighborhood, and database annotations that might introduce a potential chance for bias in the selection of functional partners. Could you kindly provide a rationale or justification for using this scoring approach?

Response. We agree with your comment about the risk of bias, but we did not consider it possible to select score evidence from certain but not all sources, as this could also introduce bias. The STRING database is a well-known and established resource, and we recognized that its statement that the scores are integrated into a final "combined score" that provides an estimate of STRING's confidence in whether a proposed association is biologically meaningful given all the contributing evidence is trustworthy. The general assumption of our study is based on the ideas of the omnigenic model, which divides genes according to their proximity in terms of causality to the phenotype, and genes that are FPs of PID genes (core genes in this study) can be treated as near-core partners of PID genes. Enrichment analysis using different approaches and databases (GO and KEGG) yielded results (Figure 1A-D, Supplementary Tables 3,4) that can be seen as indirect justification for the correctness of using combined STRING scores, as the strength of associations was greater for set I+II (score ≥0.9, closest to phenotype) than for set I+II+III (score between 0.4 and 0.89).

Thank you for pointing out the missing information about the combined STRING interaction score. We added a definition of the STRING abbreviation and explanations.

Line 124. STRING: Search Tool for the Retrieval of Interacting Gene.

Lines 126-132. Sets II and III were generated based on STRING combined scores, which provide an assessment of STRING's confidence (measured from zero to one) that the putative association between proteins is biologically significant, given all the contributing evidence from different sources. We hypothesized that the magnitude of the effect of FPs would depend on the confidence of their interaction with PID genes, with the greater the confidence, the greater the effect.

Reviewer 2 Report

Overall this is an interesting paper testing whether rare variants in primary immunodeficiency (PID) genes and their functional partners (FP) confirm risk of severe COVID-19 disease in a small cohort of young adults. The study population was selected for young age and patients in whom TREC/KREC analysis had been performed.

The authors compiled a list of genes associated with PID and their functional partners (FPs) from protein:protein interaction networks. Whole exome sequence analysis of COVID-19 patients was analysed for rare high impact and potentially pathogenic missense variants in these genes.

The major drawback of this study is the small sample size – only 9 patients with severe COVID and 11 patients with mild to moderate COVID- which reduces confidence in the results. However, I think that the approach to analysis of rare variants plus FPs is interesting and the methodology would be of use to test in a larger group of patients.

Major points

There are multiple points where the text is unclear in meaning which undermines the results presented– often because the which patients/which gene sets the analysis are being done for is not clearly explained. In particular, I cannot understand what the results In Figure 2 display from reading the text in Lines 298-312. This section needs better explanation which clearly states which patient and/or gene are being examined for each element of the figure. For example:

-where the authors state that ‘the total number of rare HI variants in PID genes was small’ it is not clear if this is in the whole group of 20 patients’

- Line 300: what does ‘these variants’ refer to – any HI variants in any gene or something else?

-Line 302; does rare variant mean HI rare variant or any rare variant?

-Line 302: where is data shown to support the text that ‘the burden of rare variants in severe versus non-302 severe COVID-19 decreased in the sets I+II > I+II+III > all genes’ 

-Line 304; what list of genes were ‘identified in the theoretical phase of the study as potentially important in associa-305 tion with COVID-19 phenotypes’?

-Line 307: what is meant by ‘obtained earlier’

-Line 324: what is meant by ‘adjusting for multiplicity’

-Line 334: I do not see this data in Figure 2D.

-Supplementary Table 6 is entitled ‘Gene annotations’ – but it is not clear if from exome sequences of the 20 patients

-Figure 2C: y axis legend – is this number of high impact variants per patient? Using which gene sets for the analysis (eg I+II+III)?

Similarly for Figure 1, I cannot understand the data shown from the legend written.

For example:

-what is the following text referring to in the Figure?

‘The legend panel for gene sets includes a built-in Venn diagram showing the overlap 258 between these sets. The analysis was performed for genes (and their partners) representing similar 259 phenotypes, i.e., belonging to the same subtables from the IUIS tables. For sets II and III, only the 260 terms encountered for the PID genes are indicated. Dotted lines separate enriched terms for genes 261 (and their partners) from each of the IUIS categories, which are signed with Arabic numerals above 262 the line graphs. See Supplementary Tables 3 and 4 for more information.’

-Figure 1 C-D: I do not understand what data these word clouds represent – are these terms enriched in biological processes and metabolic pathway gene lists? If so why is that relevant to show and  why data is displayed only for biological processes and metabolic pathway genes?

Other minor points

Materials and methods: Please explain the significance of combined interaction scores >/-= 0.9 compared with 0.4-0.89.

Were ethnicities of the patients similar between patients with severe/mild-moderate COVID-19?

Were any patients in the groups in the same family?

Line 230: do you mean IUIS sub-tables?

Line 236: what do you mean by ‘biologically important’?

Line 403: ‘within some groups of biologically important genes’ – specify which groups.

The quality of the English language is good but the level of detail of explanation for the results needs to be improved for the reader to understand them.

Author Response

We are grateful to the Reviewer for careful reading, important considerations and constructive comments. We have done our best to answer all questions. Changes in the text of the manuscript are highlighted in yellow. The line numbers refer to the current (revised) version of the manuscript.

General comment. The major drawback of this study is the small sample size – only 9 patients with severe COVID and 11 patients with mild to moderate COVID- which reduces confidence in the results. However, I think that the approach to analysis of rare variants plus FPs is interesting and the methodology would be of use to test in a larger group of patients.

Response.  Thank you. We are currently collecting a sample of critically ill patients and will continue to work on the role of the genetic landscape in the progression of acute infections. We hope to collect a larger sample for future studies and to validate the approach in a larger group of patients.

Comment 1. There are multiple points where the text is unclear in meaning which undermines the results presented– often because the which patients/which gene sets the analysis are being done for is not clearly explained. In particular, I cannot understand what the results In Figure 2 display from reading the text in Lines 298-312. This section needs better explanation which clearly states which patient and/or gene are being examined for each element of the figure. For example:

1.1 -where the authors state that ‘the total number of rare HI variants in PID genes was small’ it is not clear if this is in the whole group of 20 patients’

Response. This item is clarified. Line 332. “In the entire sample of 20 patients…”

1.2 - Line 300: what does ‘these variants’ refer to – any HI variants in any gene or something else?

Response. By "these variants" we mean rare HI variants, as these were the ones specified as the object of study at the beginning of this sentence. The number of rare HI variants and the number of genes with these variants in the entire sample of 20 patients are presented in Figure 2D (right column "Variants/Genes,#"). This clarification has been added to the legend of Figure 2. We considered all genes with the indicated variants in the entire sample as well as in gene sets I, II, and III (Methods section, lines 125-133); gene sets are indicated in the Figure (E,D), left column. Within the total gene set and sets I-III, we have indicated gene groups that may be biologically important in relation to the development and course of acute infection (Methods section, lines 136-143). These gene groups are indicated within the Y-axis breaks of the Figure 2D.

Line 333-336 “…so our primary analysis (without dividing into specific gene groups) involved comparing the burden of rare HI variants in all genes and in PID+FP genes (sets I+II and I+II+III).

Lines 355-357. The right column indicates the number of rare variants (D: HI variants; E: missense variants) and the number of genes with these variants in the entire sample of 20 patients.

1.3 -Line 302; does rare variant mean HI rare variant or any rare variant?

Response. Thank you, this is corrected. Line 334, “…the burden of rare HI variants”.

1.4-Line 302: where is data shown to support the text that ‘the burden of rare variants in severe versus non-302 severe COVID-19 decreased in the sets I+II > I+II+III > all genes’.

Response. This statement follows from the ORs reported in Figure 2 (forest plot and column 'OR (95% CI)'). This is a standard approach to interpreting results in the field (see [17, 55-59]).

We have added this clarification. Line 337. “…the burden of rare HI variants measured by OR (95% CI)…”.

1.5 -Line 304; what list of genes were ‘identified in the theoretical phase of the study as potentially important in association with COVID-19 phenotypes’?

Response. We have corrected this sentence.

Lines 339-340. performed a similar analysis for intolerant to LoF variants, haploinsufficient, essential, and immune tissue-specific genes”.

1.6. -Line 307: what is meant by ‘obtained earlier’.

Response. Thank you, we have clarified.

Line 342. “…that obtained earlier for all genes (I+II > I+II+III >all genes)”.

1.7 -Line 324: what is meant by ‘adjusting for multiplicity’

Response. Thank you for your comment. We apologize for not mentioning the threshold in the legend of Figure 2. This has been corrected. We have also corrected the wording.

Line 357-358. The p-value threshold corresponds to 0.0025 to account for multiple testing (0.05/20 comparisons).

Lines 361-362. “…the results were non-significant when corrected for multiple comparisons”.

1.8 -Line 334: I do not see this data in Figure 2D.

Response. The numbers of 22, 15 and 32 genes are given in the right column of Figure 2D 'Variants/Genes,#'. The numbers of the PID genes can be seen in Supplementary Table 6, which is mentioned at the end of the sentence.

1.9 -Supplementary Table 6 is entitled ‘Gene annotations’ – but it is not clear if from exome sequences of the 20 patients

Response. Thank you, this is corrected. “Table S6. Gene annotations for the entire gene set with variants identified by whole exome sequencing in the 20-patient sample”.

1.10 -Figure 2C: y axis legend – is this number of high impact variants per patient? Using which gene sets for the analysis (eg I+II+III)?

Response. Yes, y axis legend shows the number of rare high impact variants per patient. Y axis legend in Figure 2C has been corrected. Gene sets (I, II, III) are defined in the legend between panels D and E (highlighted in light gray).

Comment 2. Similarly for Figure 1, I cannot understand the data shown from the legend written.

2.1 -what is the following text referring to in the Figure? ‘The legend panel for gene sets includes a built-in Venn diagram showing the overlap between these sets. The analysis was performed for genes (and their partners) representing similar phenotypes, i.e., belonging to the same subtables from the IUIS tables. For sets II and III, only the terms encountered for the PID genes are indicated. Dotted lines separate enriched terms for genes (and their partners) from each of the IUIS categories, which are signed with Arabic numerals above the line graphs. See Supplementary Tables 3 and 4 for more information.’

Response. Gene sets are indicated within this legend, and the Venn diagram shows in parentheses the number of genes within each set (signed close to each element of the Venn diagram). We added this clarification (lines 279-280). The IUIS reports categorize phenotypes into tables, which in turn consist of subtables with overlapping phenotypes (lines 234-235). Because the number of subtables is large, we did not list them all in the body of the manuscript; they are listed in Supplementary Table 1. We have added missing information.

Lines 281-282. A list of subtables is provided in Supplementary Table 1.

We sequentially examined gene sets (I, I+II, and I+II+III) from all subtables to compare enrichment results using Gene Ontology and KEGG pathway enrichment analyses. Because gene sets I+II and I+II+III are significantly larger than set I, the enrichment results for sets I+II and I+II+III included terms that were missing for set I. Therefore, terms missing for set I are not shown in Figures 1A,B and Supplementary Tables 3,4, but the enrichment analysis results (strength, false discovery rate) are given according to the STRING enrichment analysis for the full set of associated terms.

We have made this clarification in the titles of Supplementary Tables 3 and 4. Terms missing for set I are not shown, but the enrichment analysis results (strength and false discovery rate) are given according to the STRING enrichment analysis for the full set of associated terms.

To show the uniform pattern of gene enrichment results from sets I, I+II, and I+II+III belonging to all IUIS categories/tables (n=9, lines 121-123), table numbers (Arabic numerals) are indicated in Figures 1A,B.

2.2 -Figure 1 C-D: I do not understand what data these word clouds represent – are these terms enriched in biological processes and metabolic pathway gene lists? If so why is that relevant to show and  why data is displayed only for biological processes and metabolic pathway genes?

Response. Thank you. Yes, word clouds represent terms enriched in biological processes and metabolic pathway gene lists. The general assumption of the study is based on the ideas of the omnigenic model, which divides genes according to their proximity in terms of causality to the phenotype, and genes that are FPs of PID genes (core genes in this study) can be treated as near-core partners of PID genes. Based on the general idea of the study, we wanted to see for ourselves and show the reader that different approaches and databases (GO and KEGG) give the main results that are similar. This is an indirect justification for the results of using STRING combined scores with different cut-offs, since the strength of the associations is greater for set I+II (score ≥0.9, closest to the phenotype) than for set I+II+III (score 0.4 to 0.89).

            GO and KEGG pathway enrichment analyses are among the most widely used and validated types of enrichment analyses based on functional annotations of genes according to their participation in biological processes and pathways, covering the main types of functional annotations important in the context of our work.

Other minor points

Comment 3. Materials and methods: Please explain the significance of combined interaction scores >/-= 0.9 compared with 0.4-0.89.

Response. We added this explanation.

Lines 130-132. We hypothesized that the magnitude of the effect of FPs would depend on the confidence of their interaction with PID genes, with the greater the confidence, the greater the effect.

Comment 4. Were ethnicities of the patients similar between patients with severe/mild-moderate COVID-19?

Response. Yes. Principal component analysis (PCA) was performed on the entire cohort of 86 patients [17]. 20 patients from this cohort who were included in the current study were similar in terms of ethnicity.

Comment 5. Were any patients in the groups in the same family?

Response. There were no relatives among patients.

Comment 6. Line 230: do you mean IUIS sub-tables?

Response. Thank you, yes. Line 249, ‘IUIS’ is added.

Comment 7. Line 236: what do you mean by ‘biologically important’?

Response. There are many different metrics, from which we selected some basic metrics such as haploinsufficiency, essentiality for life, and intolerance to variants with predicted pathogenicity. Given study specificity, we also included genes associated with SARS-CoV-2 infection and/or COVID-19 disease from the GENCODE project and immune tissue-specific. See our response to your comment 1.5. The section describing these gene groups is referenced.

Line 256. “…the proportion of biologically important genes (see section 2.2)…”.

Comment 8. Line 403: ‘within some groups of biologically important genes’ – specify which groups.

Response. Thank you, this is done.

Line 447. “…(intolerant to LoF variants, haploinsufficient and essential for life)

Comments on the Quality of English Language

The quality of the English language is good but the level of detail of explanation for the results needs to be improved for the reader to understand them.

We would like to reiterate our appreciation to the Reviewer for his/her careful consideration of our paper. We have tried to make the revision exactly according to the questions and comments, and hope that the revised version of the paper is much clearer and better presented.